# Structural basis for activation and gating of IP$_3$ receptors

Emily A. Schmitz [1,2,3], Hirohide Takahashi [1,2,3] & Erkan Karakas [1,2✉]

A pivotal component of the calcium (Ca$^{2+}$) signaling toolbox in cells is the inositol 1,4,5-triphosphate (IP$_3$) receptor (IP$_3$R), which mediates Ca$^{2+}$ release from the endoplasmic reticulum (ER), controlling cytoplasmic and organellar Ca$^{2+}$ concentrations. IP$_3$Rs are co-activated by IP$_3$ and Ca$^{2+}$, inhibited by Ca$^{2+}$ at high concentrations, and potentiated by ATP. However, the underlying molecular mechanisms are unclear. Here we report cryo-electron microscopy (cryo-EM) structures of human type-3 IP$_3$R obtained from a single dataset in multiple gating conformations: IP$_3$-ATP bound pre-active states with closed channels, IP$_3$-ATP-Ca$^{2+}$ bound active state with an open channel, and IP$_3$-ATP-Ca$^{2+}$ bound inactive state with a closed channel. The structures demonstrate how IP$_3$-induced conformational changes prime the receptor for activation by Ca$^{2+}$, how Ca$^{2+}$ binding leads to channel opening, and how ATP modulates the activity, providing insights into the long-sought questions regarding the molecular mechanism underpinning receptor activation and gating.

---

[1] Department of Molecular Physiology and Biophysics, Vanderbilt University, School of Medicine, Nashville, TN 37232, USA. [2] Center for Structural Biology, Vanderbilt University, Nashville, TN 37232, USA. [3] These authors contributed equally: Emily A. Schmitz, Hirohide Takahashi. ✉email: erkan.karakas@vanderbilt.edu

P$_3$Rs are intracellular Ca$^{2+}$ channels, predominantly localized to the ER and activated by the binding of IP$_3$ generated in response to external stimulation of G-protein coupled receptors[1–3]. Opening of the IP$_3$Rs results in the rapid release of Ca$^{2+}$ from the ER lumen into the cytoplasm, triggering diverse signaling cascades that regulate physiological processes such as learning, fertilization, gene expression, and apoptosis. Dysfunctional IP$_3$Rs cause abnormal Ca$^{2+}$ signaling and are associated with many diseases, including diabetes, cancer, and neurological disorders[4,5]. There are three IP$_3$R subtypes (IP$_3$R-1, -2, and -3) that share 60–70% sequence identity, form homo- or hetero-tetramers, exhibit different spatial expression profiles, and are involved in different signaling pathways[1–3]. Each IP$_3$R subunit is about 2700 amino acids in length and contains a transmembrane domain (TMD) and a large cytoplasmic region comprising two β-trefoil domains (βTF1 and βTF2), three Armadillo repeat domains (ARM1, ARM2, and ARM3), a central linker domain (CLD), a juxtamembrane domain (JD), and a short C-terminal domain (CTD)[6–10] (Fig. 1).

In addition to IP$_3$, the receptor activation requires Ca$^{2+}$ at nanomolar concentrations, whereas Ca$^{2+}$ at higher concentrations is inhibitory, causing the receptor to be tightly regulated by Ca$^{2+}$,[11–15]. The cryo-EM structure of human IP$_3$R-3 (hIP$_3$R-3) in the presence of the inhibitory Ca$^{2+}$ concentrations (2 mM) revealed two binding sites[6]. However, their role in channel activation and inhibition has remained uncertain. Furthermore, although ATP binding potentiates the receptor by increasing the open probability and duration of the channel openings, the underlying molecular mechanism has not been uncovered[16,17]. In this study, we illuminate the structural framework of receptor activation and channel opening by analyzing five cryo-EM

structures of hIP$_3$R-3 in the closed-pre-activated, open-activated, and closed-inactivated conformations.

## Results and discussion

**Cryo-EM structures of hIP$_3$R-3 gating conformations.** Bimodal regulation of IP$_3$R activity by Ca$^{2+}$ complicates sample preparation because of the requirement for fine adjustment of Ca$^{2+}$ concentration to trap the channel in the open conformation. Although free Ca$^{2+}$ concentrations in solutions can easily be controlled by using Ca$^{2+}$ buffers such as EGTA or BAPTA, it becomes challenging during sample preparation for cryo-EM due to the small volumes used. Typically, a 2–3 μl protein sample is applied to a cryo-grid, but more than 99% of the sample volume is lost during grid preparation due to extensive blotting with filter paper[18]. During this time, the samples contact filter paper and cryo-grids, containing various amounts of Ca$^{2+}$. Small sample volumes and short time frames may reduce these buffers' efficiency, causing the free Ca$^{2+}$ concentration to increase to inhibitory levels prior to sample freezing. In order to maximize the chances of obtaining particles in the active state, we prepared the sample in: (1) EDTA, which has ~200 fold faster binding kinetics to Ca$^{2+}$ than EGTA[19], a common Ca$^{2+}$ chelator, and is more likely to chelate excess Ca$^{2+}$ and other divalent cations within the short period prior to the sample plunging, (2) ATP, which increases the open probability of IP$_3$Rs and dampens the inhibitory effect of Ca$^{2+}$,[16,17], and (3) high concentrations of IP$_3$.

The final hIP$_3$R-3 sample was purified in the presence of 1 mM EDTA and supplemented with 0.5 mM IP$_3$, 5 mM ATP, and 0.1 mM CaCl$_2$ before preparing cryo-grids. Although the free Ca$^{2+}$ concentration was calculated around 100 nM under these conditions using Maxchelator[20], the actual free Ca$^{2+}$

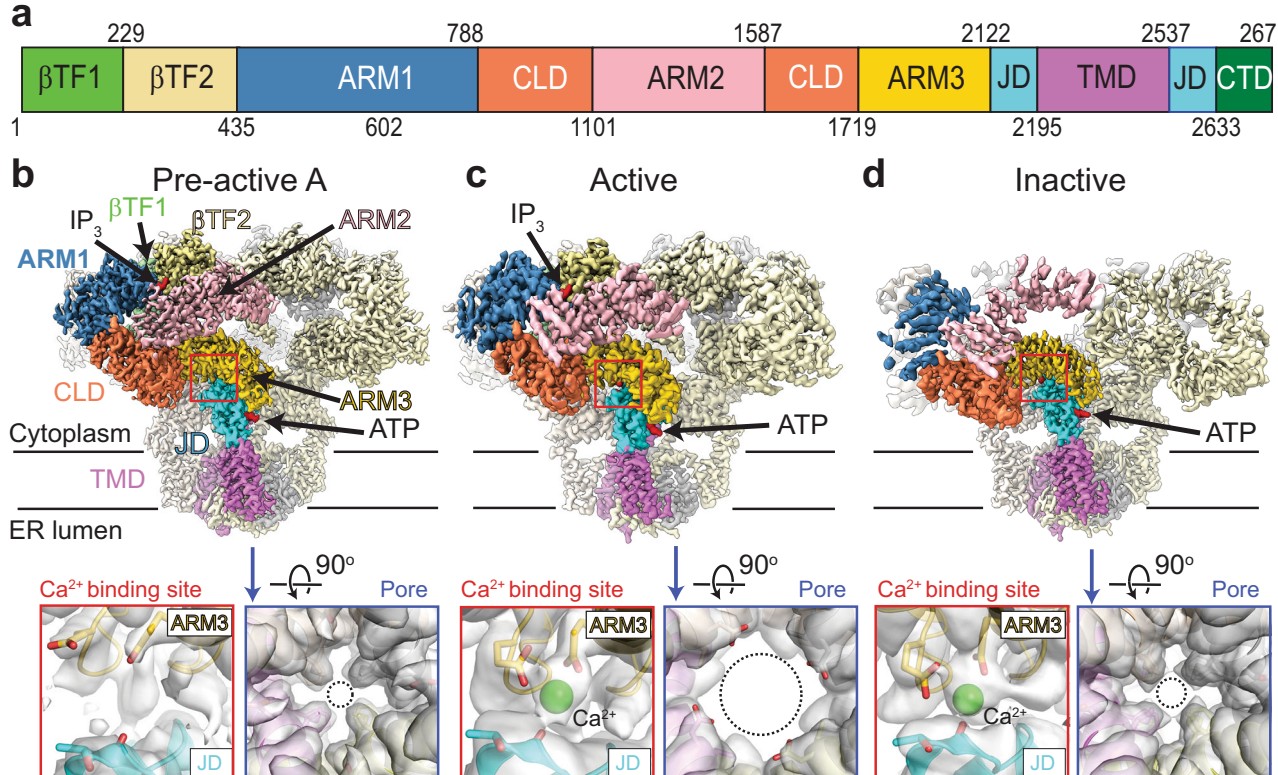

**Fig. 1 Cryo-EM structures of hIP$_3$R-3 in multiple conformations. a** Domain boundaries of hIP$_3$R-3. **b–d** Composite maps of hIP$_3$R-3 in pre-active A (**b**), active (**c**), inactive (**d**) conformations. Each domain in one of the subunits is colored as in (**a**). Maps within the boxes, shown transparent, are close-up views of the Ca$^{2+}$ binding site (red) and the pore (blue) with ribbon representation of hIP$_3$R-3. Select residues are shown in the sticks. Dashed circles indicate opening through the gate.

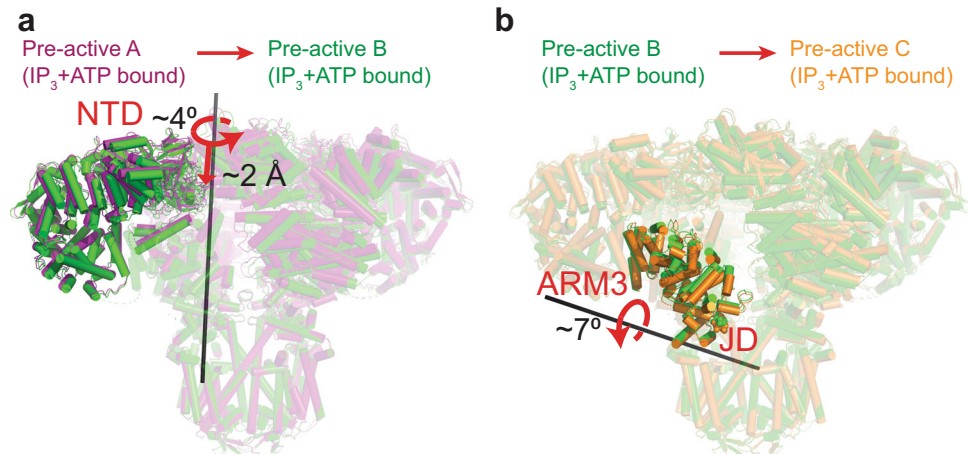

**Fig. 2 Conformational changes in the pre-active states. a, b** Ribbon representations of hIP$_3$R structures superposed on the residues forming the selectivity filter and P-helix of the TMDs, emphasizing the conformational changes between the states indicated above. Domains with substantial conformational changes are shown in full colors only on one subunit, while the rest of the protein is transparent. Curved and straight red arrows indicate the rotation and translation of the domains with red labels relative to the rotation axis (black bars), respectively.

concentration may be higher due to potential leakage of Ca$^{2+}$ during the cryo-grid preparation as mentioned above. We performed a cryo-EM analysis on a large dataset by employing exhaustive 3D classification strategies to separate particles belonging to different functional states resulting in five high resolution (3.2–3.8 Å) structures (Supplementary Figs. 1–6; Supplementary Table 1). The pore region in all structures resolved to 3.5 Å or better, allowing us to build side chains and determine if the channel was open or closed (Fig. 1; Supplementary Figs. 2–6).

Three structures have closed pores with well-resolved densities for IP$_3$ and ATP and are referred to as pre-active A, B, and C (Fig. 1b; Supplementary Figs. 1–7). The structure named "active" displays drastic conformational changes at the TMD, leading to pore opening (Fig. 1c). In addition to the well-resolved densities for IP$_3$ and ATP, the active structure reveals substantial density, interpreted as Ca$^{2+}$, at the ARM3-JD interface, referred to as the activatory Ca$^{2+}$ binding site (Fig. 1c; Supplementary Figs. 5, 7). In the fifth structure, the channel is closed, the activatory Ca$^{2+}$ binding site is occupied, and the intersubunit interactions of the cytoplasmic domains are lost (Fig. 1d; Supplementary Fig. 6). The structure is highly similar to the hIP$_3$R-3 structures obtained in the presence of inhibitory Ca$^{2+}$ concentrations[6], except for βTF1, which moves closer to the ARM1 (Supplementary Fig. 8). Most notably, ARM2 adopted the same conformation relative to ARM1 and CLD, creating the binding site for the second Ca$^{2+}$ observed at high Ca$^{2+}$ concentrations (Supplementary Fig. 8). While these similarities suggest a Ca$^{2+}$ ion occupies this site and the structure represents the Ca$^{2+}$ inhibited state, the quality of the map around the region did not allow accurate inspection of the presence of Ca$^{2+}$ (Supplementary Fig. 8). Therefore, we refer to the structure as "inactive" while it remains unclear if it represents a desensitized state that hIP$_3$R-3 adopts without additional Ca$^{2+}$ binding or an inhibited state forced by binding of additional Ca$^{2+}$ to an inhibitory site.

It is important to note that our initial 3D classification runs resulted in two major classes grouping the pre-active and active structures into one class and the inactive structure into another (Supplementary Fig. 1). It was essential to perform another round of 3D classification focusing only on the core of the protein to separate the particles in the active state from the pre-active states, potentially due to subtle differences in the overall structures and the much fewer number of particles in the active state (20,039 particles compared to 346,684 particles in the pre-active states) (Supplementary Fig. 1; Supplementary Table 1).

**Priming of hIP$_3$R-3 for activation.** To compare the structures presented here, we aligned their selectivity filters and pore helices (residues 2460-2481), which reside at the luminal side of the TMD and are virtually identical in all classes. The pre-active A structure is almost identical to the previously published IP$_3$-bound hIP$_3$R-3 structure[6] (Supplementary Fig. 9a) and reveals that IP$_3$ binding causes the ARM1 to rotate about 23° relative to the βTF-2, causing global conformational changes within the cytoplasmic domains, as observed in previous cryo-EM and X-ray crystallography experiments[6,8,21–24] (Supplementary Fig. 9b, c). The pre-active B and C structures adopt distinct conformations that are intermediates between the pre-active A and open state structures. Based on these conformational changes, we propose a sequential transition from pre-active A to B, then C, although the alternative transitions cannot be ruled out entirely. During the transition to the pre-active B state, the N-terminal domain (NTD) of each protomer comprising βTF1, βTF2, ARM1, ARM2, and CLD rotates about 4° counter-clockwise relative to the TMD and moves about 2 Å closer to the membrane plane (Fig. 2a; Supplementary Movie 1). In the pre-active C state, the NTDs remain primarily unchanged compared to the pre-active B state, while the ARM3 and JD are rotated by 7°, causing mild distortions at the cytoplasmic side of the TMD without opening the channel (Fig. 2b; Supplementary Movie 1). Compared to the ligand-free conformation, the βTFs move about 7 Å closer to the membrane plane, and ARM3-JD rotates about 11° in the pre-active C conformation.

**Ca$^{2+}$-mediated conformational changes leading to pore opening.** In the absence of Ca$^{2+}$, the ARM3 and JD act as a rigid body, where there are no significant conformational changes relative to each other (Fig. 2a, b). When Ca$^{2+}$ is bound, the JD rotates about 11° relative to the ARM3 (Fig. 3a), resembling a clamshell closure, which leads to global conformational changes in the whole receptor, including the movement of the NTD closer to the membrane plane by 2 Å (Fig. 3b; Supplementary Movie 1). In contrast to the limited rotation of the ARM3 (about 5°), the JD rotates about 14° around an axis roughly perpendicular to the membrane plane, leading to conformational changes at the TMD and resulting in pore opening in the active state (Fig. 3b).

Ca$^{2+}$ is coordinated by E1882 and E1946 on the ARM3 and the main-chain carboxyl group of T2581 on the JD (Fig. 3c). H1884 and Q1949 are also close and may interact with Ca$^{2+}$ through

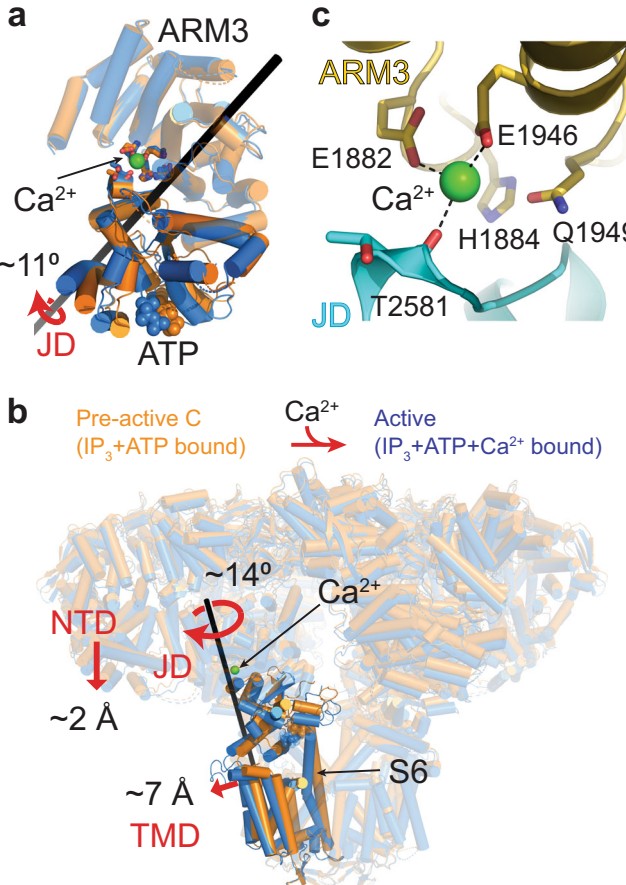

**Fig. 3 Conformational changes coupling Ca²⁺ binding to pore opening.**
**a** Comparison of the JD (shown in full colors) orientation relative to the ARM3 (shown transparent) in the pre-active-C (orange) and active (blue) structures. The black bar indicates the axis for the rotation of the JD. Ca²⁺ and ATP are shown as spheres. **b** Global conformational changes induced by Ca²⁺ binding are depicted similar to Fig. 2. **c** Close-up view of the Ca²⁺ binding site in the active conformation. Domains are colored as in Fig. 1.

water molecules (Fig. 3c). These residues are highly conserved in the homologous ion channel family, ryanodine receptors (RyRs), suggesting a common activation mechanism in IP₃Rs and RyRs[25] (Supplementary Fig. 10a, b). Mutation of the corresponding residues in RyRs markedly reduced the sensitivity to Ca²⁺, further supporting this site's involvement in the Ca²⁺ induced activation[26–28].

**ATP binding site.** Within the JD, we observed a well-resolved cryo-EM density for ATP in all the structures (Supplementary Fig. 7). The quality of the maps obtained through local refinement allowed unambiguous modeling of ATP, revealing its key interactions with the protein residues (Fig. 4a, b). The adenosine base intercalates into a cavity surrounded by F2156, F2539, I2559, M2565, and W2566 near the zinc finger motif and forms hydrogen bonds with the sulfur of C2538, the backbone amide group of F2539, and the carbonyl groups of H2563 and I2559 (Fig. 4b). The phosphate moieties interact with K2152, K2560, and N2564 (Fig. 4b). There are no apparent structural changes around the binding site upon ATP binding, suggesting that ATP's potentiating effect is likely due to the increased rigidity of the JD (Supplementary Fig. 9d). ATP binding site is highly conserved among the subtypes except for E2149 which corresponds to lysine and arginine in IP₃R-1 and IP₃R-2 (Fig. 4c). A positively charged residue instead of E2149 in the proximity of the

phosphate moieties may cause tighter interaction of ATP with IP₃R-1 and IP₃R-2, explaining the low binding affinity of ATP to IP₃R-3 compared to IP₃R-1 and IP₃R-2[17].

ATP binds to a similar location near the zinc finger motif in RyRs[25,29,30]. However, its binding mode differs, potentially due to the differences in the residues that form the binding pocket, most notably the basic residues interacting with the phosphate moieties (Supplementary Fig. 10a, c, d). In RyR-1s, the phosphate moieties interact with K4211, K4214, and R4215, all located on a single helix (Supplementary Fig. 10c, d). In hIP₃R-3, there is only a single lysine residue (K2152) on the corresponding helix, and the phosphate moieties interact with K2560, located on the opposite side of the binding pocket. A leucine residue (L4980) occupies this position in RyR-1s. The differences in the number and location of the basic residues likely force the phosphate moieties of ATP to adopt different conformations. Furthermore, F2156 in hIP₃R-3 points toward the adenosine binding pocket, prohibiting ATP from adopting the conformations observed in RyR-1s due to steric clash in hIP₃R-3s (Supplementary Fig. 10c, d).

**Structure of the TMD in the open conformation.** The TMD of IP₃Rs has the same overall architecture of voltage-gated ions channels with a central pore domain, consisting of S5, S6, and pore (P) helix, surrounded by pseudo-voltage-sensor domains (pVSDs), consisting of S1, S2, S3, and S4 helices along with two IP₃R/RyR specific TM helices (S1' and S1") (Fig. 5). In the closed channel, F2513 and I2517 of the S6 helix form two layers of hydrophobic constriction at the pore, blocking the path for the permeation of hydrated ions (Fig. 5). JD's rotation upon Ca²⁺ binding pushes the pVSD's cytoplasmic side away from the pore domain by about 7 Å and tilts the cytoplasmic side of the S6 (S6_cyt) by 12° (Fig. 5a; Supplementary Movie 1). Concurrently, the S4-5 linker and S5 helix move away from the S6 helix, thereby inducing a distortion of S6 around the constriction site and moving F2513 and I2517 away from the pore. As a result, the diameter of the water-accessible pore increases to 8 Å, large enough to permeate hydrated cations (Fig. 5b). The flexibility introduced by the neighboring glycine residue (G2514), mutation of which to alanine in IP₃R-1 is associated with spinocerebellar ataxia 29 (SCA29)[31], is likely critical to the movement of F2513. The tilting of the S6_cyt breaks the salt bridge between D2518 and R2524 of the neighboring subunits, moving D2518 towards the pore while pulling R2524 away, which creates an electronegative path on the cytoplasmic side of the pore (Supplementary Fig. 11). In contrast to the prediction of a π- to α-helix transition at the S6_lum during channel opening[6,10], the π-helix remains intact, and its tip acts as a pivot for the S6_cyt tilting and bulging (Fig. 5).

Although the TMDs of IP₃Rs and RyRs are highly similar, there are noticeable differences in their pore structures (Supplementary Fig. 10e, f). In RyRs, the constriction site is formed by glutamine and isoleucine, corresponding to F2513 and I2517 in hIP₃R-3, respectively[32]. In the open state of RyRs, the isoleucine is positioned similarly to I2517 of hIP₃R-3[25,33,34]. On the other hand, the glutamine residue faces the pore in the open state, forming part of the hydrophilic permeation pathway, unlike F2513. Interestingly, N2510 in hIP₃R-3, which corresponds to alanine in RyRs, faces the permeation pathway similar to the glutamine of RyRs, suggesting that the amide group plays an important role in the ion permeation. However, since the side chain of N2510 extends from a different position on the S6 helix than the side chain of glutamine in RyRs, the binding pocket for ryanodine[25], a RyR-specific inhibitor, is not present in IP₃R, potentially causing IP₃Rs to be unresponsive to ryanodine[32].

Several missense mutations identified in the IP₃R subtypes are associated with diseases, including spinocereblar ataxia, Gillespie

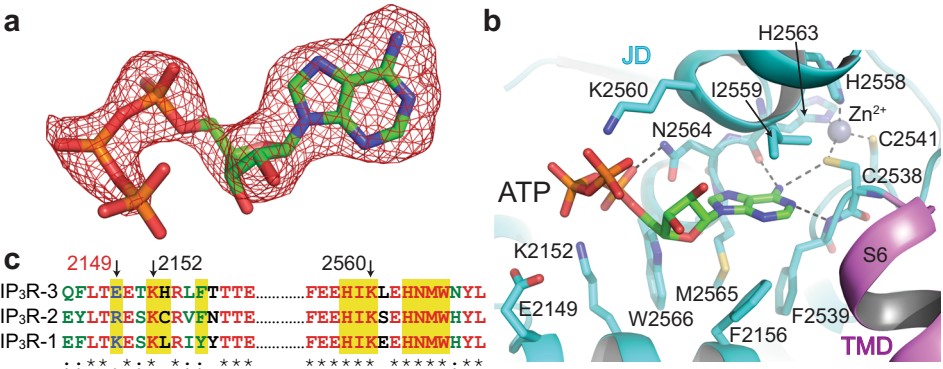

**Fig. 4 ATP binds to the JD. a** The cryo-EM density of ATP (red mesh) from the composite map of the pre-active A state and the modeled ATP molecule. **b** Close-up view of the ATP binding site in the pre-active A state. Dashed lines indicate hydrogen bonding. **c** Sequence alignment of hIP₃R subtypes around the residues forming the ATP binding site. Residues shown in (**b**) are highlighted. Select residues are indicated by arrows, and E2149 is labeled in red.

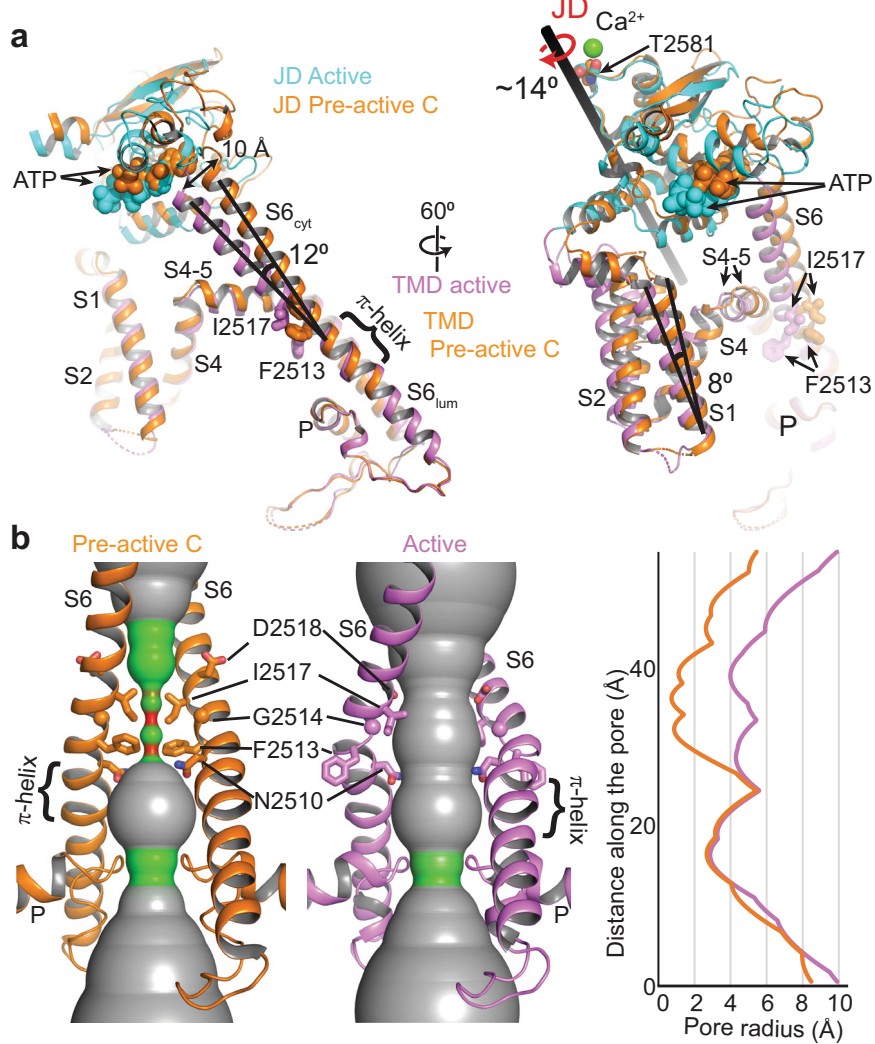

**Fig. 5 Structure of the IP₃R-3 in the open conformation. a** Comparison of the hIP₃R-3 structures in the pre-active C and active conformations aligned as in Fig. 2. **b** Ion permeation pathways of hIP₃R-3 in pre-active C and active conformations (radii coloring: red, <0.8 Å; green, 0.8-4.0 Å; gray, >4.0 Å) along with the 1D graph of the pore radius. The JD and TMD of the active state are shown in cyan and violet, respectively. The pre-active C structure is colored orange.

syndrome, anhidrosis, and neck squamous cell carcinoma (Supplementary Fig. 12; reviewed in[32,35,36]). Perhaps not surprisingly, most of these mutations are localized around the IP$_3$ binding site and alter IP$_3$ binding affinity[32,35–37]. Another hot spot for these mutations is the constriction site of the pore, which undergoes conformational changes during channel opening (Supplementary Fig. 12). It is plausible that these mutations either affect the Ca$^{2+}$ permeability (e.g., mutation of N2510[38] or I2517[39]) or restrict conformational changes required for dilation of the pore (e.g., mutation of G2514[31]). Two of the mutated residues (T2519[40] and F2520[41]) interact with the residues on the S4-S5 linker, which couples the tilting of the pVSD to the bulging of the constriction site (Supplementary Fig. 12b). Mutations of these residues are likely to impair this coupling and thus hinder gating.

**The flexibility of the CTD.** The CTD, extending from the JD along the symmetry axis, forms a left-handed coiled-coil motif before interacting with the βTF2 of the neighboring subunit. The density for the CTD was poorly resolved in all of the states (Supplementary Fig. 13a, b). However, the coiled-coil motifs were visible in the unsharpened maps in the pre-active and active states, enabling us to model poly-alanine peptides without assigned registries (Supplementary Fig. 13a, b). The densities for the extensions from the coiled-coil motif towards the βTF2 become visible when viewed at lower thresholds, whereas the linkers between the JD and the coiled-coil motif remain invisible, indicating higher flexibility for this region (Supplementary Fig. 13a, b). We did not observe any interpretable density for the CTD in the inactive state (Supplementary Fig. 13a, b).

For IP$_3$R-1, the CTD was proposed to transmit the conformational changes induced by IP$_3$ at the NTD to the JD[8]. In IP$_3$R-3, there are no apparent changes on the coiled-coil motif in the pre-active states, but the coiled-coil motif rotates about 20° around the symmetry axis and moves closer to the TMD by 6 Å in the active state (Supplementary Fig. 13c, d). However, the linker between the coiled-coil motif and JD remains flexible, suggesting that the structural rearrangements of this domain are not directly enforcing the channel opening (Supplementary Fig. 13). In line with these observations, removing CTD residues interacting with the βTF2 or swapping the C-terminal region of IP$_3$R-1 with the RyRs, which lack the extended CTD, did not diminish receptor activation[21,24,42].

**Mechanism of hIP$_3$R-3 activation and gating.** It has been long recognized that IP$_3$ binding primes the receptor for activation by Ca$^{2+}$,[43] but how the priming is achieved has remained elusive. Our structures reveal that IP$_3$ binding leads to several conformational changes at the NTD, ARM3, and JD, without any apparent structural changes at the activatory Ca$^{2+}$ binding site, and that the ARM3 and JD adopt a new pre-gating conformation relative to the TMD with modest changes at the intersubunit interface between the JDs at the cytoplasmic side of the TMD (Fig. 6; Supplementary Fig. 14; Supplementary Movie 1). In addition, ARM3s are constrained in their pre-gating conformation by the tetrameric cage-like assembly of the NTDs, forcing the JDs to rotate upon Ca$^{2+}$ binding. The NTD assembly is maintained by the βTF1-βTF2 intersubunit interactions (βTF ring), which remains intact in the pre-active and active states (Fig. 6; Supplementary Fig. 15; Supplementary Movie 1) and acts as a pivot for the conformational changes that stabilize the ARM3. On the other hand, its disruption in the inactive state leads to the loosening of the tetrameric assembly of the NTDs, relieving the ARM3 constraints and causing the JD and TMD to adopt the closed channel conformation despite the bound Ca$^{2+}$ to the

activatory site. Supporting this hypothesis, the removal of βTF1 or mutation of W168, which resides at the βTF1-βTF2 interface (Supplementary Fig. 15), was shown to abolish IP$_3$R activity[44,45].

In conclusion, the ensemble of structures obtained from the same sample demonstrates structural heterogeneity of IP$_3$Rs in the presence of IP$_3$, ATP, and Ca$^{2+}$. Our ability to correlate these structures with their plausible functional states allowed us to define the conformational changes at different gating states, revealing the structural features that underpin IP$_3$R activation and gating. These structures will likely serve as foundations for future experiments addressing biophysical and functional questions related to IP$_3$Rs. Furthermore, our study reinforces the power of cryo-EM in analyzing heterogeneous samples and highlights the importance of a thorough investigation of the data to identify physiologically relevant conformations, even when they constitute only a tiny fraction of the sample.

## Methods

**Protein expression and purification.** Expression and purification of hIP3R-3 were performed as previously described with minor modifications[10]. Briefly, hIP3R-3 (residues 4-2671) with a C-terminal OneStrep tag was expressed using the MultiBac expression system[46]. Sf9 cells ($4 \times 10^6$ cells/mL) were harvested by centrifugation ($4000 \times g$) 48 h after infection with the baculovirus. Cells resuspended in a lysis buffer of 200 mM NaCl, 40 mM Tris-HCl pH 8.0, 2 mM EDTA pH 8.0, 10 mM β-mercaptoethanol (BME), and 1 mM Phenylmethylsulfonyl fluoride (PMSF) were lysed using Avastin EmulsiFlex-C3. After centrifugation of the lysate at $7000 \times g$ for 20 min to remove large debris, the membrane was pelleted by centrifugation at $185,000 \times g$ (Type Ti45 rotor) for 1 h. Membrane pellets were homogenized in ice-cold resuspension buffer (200 mM NaCl, 40 mM Tris-HCl pH 8.0, 2 mM EDTA pH 8.0, 10 mM BME) using a Dounce homogenizer, and solubilized using 0.5% Lauryl maltose neopentyl glycol (LMNG) and 0.1% glyco-diosgenin (GDN) at a membrane concentration of 100 mg/mL. After 4 h of gentle mixing in the cold room, the insoluble material was pelleted by centrifugation at $185,000 \times g$ (Type Ti45 rotor) for 1 h, and the supernatant was passed through Strep-XT resin (IBA Biotagnology) via gravity flow. The resin was washed first with 5 column volume (CV) of wash buffer composed of 200 mM NaCl, 20 mM Tris-HCl pH 8.0, 10 mM BME, 0.005% GDN, 0.005% LMNG, followed by 5 CV of wash buffer supplemented with 5 mM ATP and 20 mM MgCl$_2$ to remove any bound chaperone proteins, and finally with 5CV of wash buffer supplemented with 1 mM EDTA. The protein was eluted using wash buffer supplemented with 1 mM EDTA and 100 mM D-Biotin (pH 8.2). The protein was further purified by size exclusion chromatography (SEC) using a Superose 6 Increase column (10/300 GL, GE Healthcare) equilibrated with the SEC buffer composed of 200 mM NaCl, 20 mM Tris-HCl pH 8.0, 1 mM EDTA pH 8.0, 2 mM TCEP, 0.005% LMNG, and 0.005% GDN. The fractions corresponding to hIP3R-3 were combined and concentrated to 4 mg/mL using a 100 kDa centrifugal filter (Sartorius). The concentrated sample was then centrifuged at $260,000 \times g$ using an S100AT rotor (ThermoFisher Scientific). The concentration dropped to 1.8 mg/mL.

**Cryo-EM sample preparation and data collection.** Purified hIP$_3$R-3 in the SEC buffer containing 1 mM EDTA was supplemented with 500 μM IP$_3$ (from 10 mM stock in water), 0.1 mM CaCl$_2$, and 5 mM ATP (from 100 mM stock, pH 7.2). 2.0 μL of the protein sample was applied to 300 mesh Cu Quantifoil 1.2/1.3 grids (Quantifoil Microtools) that were glow discharged for 20 s at 25 mA. The grids were blotted for 7 s at force 10 using single-layer Whatman ashless filter papers (Cat. #: 1442-055, GE Healthcare) and were plunged into liquid ethane using an FEI MarkIV Vitrobot at 8 °C and 100% humidity. The filter papers were not pre-treated with Ca$^{2+}$ chelators or any other chemicals. Four grids prepared using the same sample were imaged using a 300 kV FEI Krios G3i microscope equipped with a Gatan K3 direct electron camera in four different data collection sessions at Case Western Reserve University. Movies containing 40–50 frames were collected at a magnification of ×105,000 in super-resolution mode with a physical pixel size of 0.828 Å/pixel and defocus values at a range of −0.8 to −1.6 μm using the automated imaging software SerialEM[47] and EPU (ThermoFisher Scientific).

**Cryo-EM data processing.** Datasets from four sessions were initially processed separately using Relion 3.0[48]. We used MotionCor2[49] and Gtcf[50] to perform beam-induced motion correction and CTF estimations, respectively. We performed auto picking using the Laplacian-of-Gaussian option of Relion, extracted particles binned $4 \times 4$, and performed 2D class classification. Using the class averages with apparent features, we performed another round of particle picking. We cleaned the particles, extracted as $4 \times 4$ binned, through 2D classification and performed 3D classification using the hIP3R-3 map (EMD-20849[10]), which was converted to the appropriate box and pixel size. We observed two predominant conformations. One had a compact NTD and tight interactions between subunits as in previously published IP$_3$R structures in the absence of Ca$^{2+}$, hereafter called "compact"

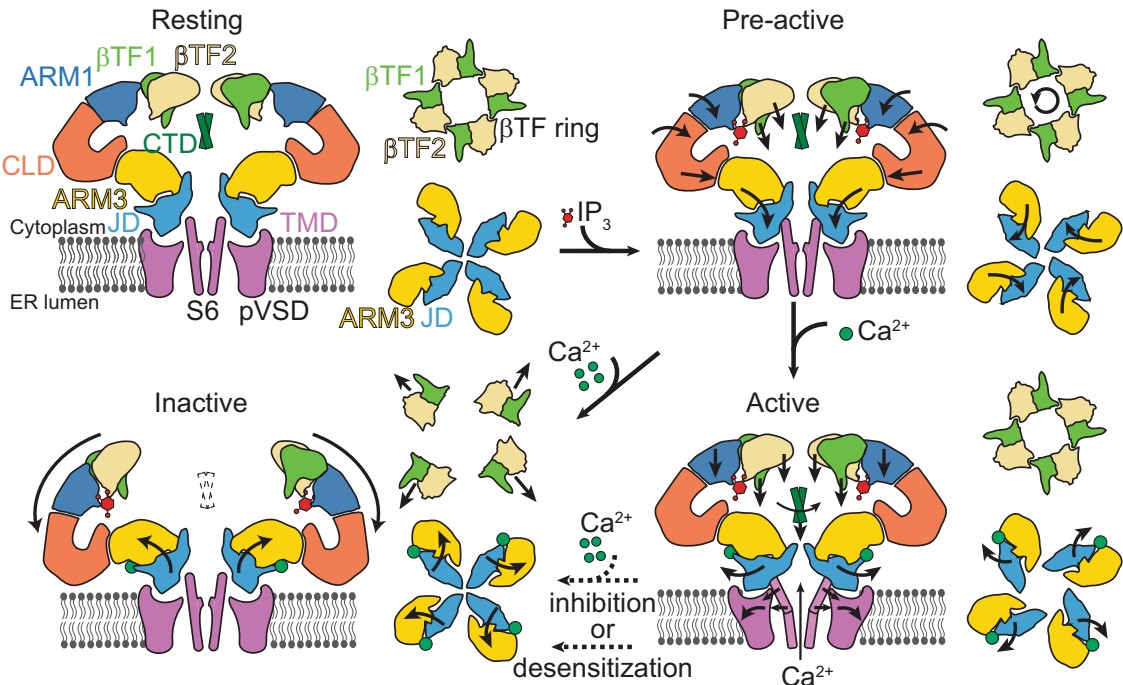

**Fig. 6 Schematic representation of the IP₃R gating cycle.** A side view of the two opposing subunits (left) and cytoplasmic views of the βTFs and ARM3-JD tetramers (right) for each indicated functional state is depicted. The ARM2 and CTD are omitted for clarity. Arrows shown on the domains indicate the direction of the rotation or translation from the previous conformation.

conformation[6–8,10] (Supplementary Fig. 1). The other one had the NTD of each subunit tilted away from the central symmetry axis resembling hIP₃R-3 structures obtained in the presence of high $Ca^{2+}$ concentrations, hereafter called "loose" conformation[6] (Supplementary Fig. 1). These particles were separately selected and reextracted using a box size of 480 × 480 pixels at the physical pixel size. After 3D refinements, we performed CTF refinement and Bayesian polishing[51].

We combined all the polished particles and performed another round of 3D classification, using one of the compact structures as a reference map. We grouped particles into "compact" and "loose" classes (Supplementary Fig. 1). Refinement of the particles in the "compact" conformation yielded a 3D reconstruction with an average resolution of 3.9 Å. Although there were slight changes at the TMD compared to the structure in the closed state, these changes were not significant enough to suggest that the channel was open. To more clearly resolve the density around the TMD, we performed another round of 3D classification using a mask that only covers the ARM3, JD, and TMD and without performing an angular or translational alignment in Relion3 (Supplementary Fig. 1)[52]. 3D refinements of the particles in each class were performed using non-uniform refinement in CryoSPARC, enforcing C4 symmetry and local CTF refinements (Supplementary Fig. 1)[53]. Five classes led to four high-resolution (better than 4 Å) 3D reconstructions, whereas the 3D refinement of the other three classes resulted in poorly resolved maps. The particles in the "loose" conformation were processed using non-uniform refinement in CryoSPARC, but without enforcing any symmetry. Local resolution estimates were calculated using CryoSPARC[53] (Supplementary Figs. 2–6). Some of the data processing and refinement software was supported by SBGrid[54].

To improve the quality of the maps, we performed local refinements using masks covering parts of the original cryo-EM maps (Supplementary Figs. 2–6). We prepared five masks that cover distinct domains of one of the subunits for the pre-active A, B, C, and active conformations. After symmetry expansion using C4 symmetry, we performed local refinement using CryoSparc (Supplementary Figs. 2–5). For the inactive state, we prepared four masks that cover the cytoplasmic domains of each subunit and another mask that covers the tetrameric ARM3, JD, and TMD (Supplementary Fig. 6). The local refinements were performed using C1 symmetry for the cytoplasmic domains and C4 symmetry for the tetrameric ARM3, JD, and TMD. The resulting local refinement maps were aligned onto the original maps using Chimera[55] and merged using the "VOP maximum" command of Chimera[55] to prepare the composite maps (Supplementary Figs. 2–6).

**Model building.** Model building was performed using Coot[56]. We first placed the hIP₃R-3 structure in ligand-free conformations (PDB ID: 6UQK[10]) into the composite map of Pre-active A, and performed rigid-body fitting of individual domains of one of the protomers. We then manually fit the residues into the density and expanded the protomer structure into a tetramer using the C4 symmetry. We performed real-space refinement using Phenix[57]. We repeated

build-refine iterations till a satisfactory model was obtained. This model was used as a starting model for the other structures following the same workflow. Regions without interpretable densities were not built into the model. Residues without apparent density for their side chains were built without their side chains (i.e., as alanines) while maintaining their correct labeling for the amino acid type. The coiled-coil regions were modeled as poly-alanines without residue assignment using the unsharpened maps. Validations of the structural models were performed using MolProbity[58] implemented in Phenix[57].

**Figure preparation.** Figures were prepared using Chimera[55], ChimeraX[59], and The PyMOL Molecular Graphics System (Version 2.0, Schrödinger, LLC). Calculation of the pore radii was performed using the software HOLE[60].

**Reporting summary.** Further information on research design is available in the Nature Research Reporting Summary linked to this article.

## Data availability

The data that support this study are available from the corresponding author upon reasonable request. Cryo-EM maps and atomic coordinates are deposited to the Electron Microscopy Data Bank (EMDB) and Protein Data Bank (PDB) databases, respectively. The accession codes are EMD-25667 and 7T3P for pre-active A, EMD-25668 and 7T3Q for pre-active B, EMD-25669 and 7T3R for pre-active C, EMD-25670 and 7T3T for active, and EMD-25671 and 7T3U for inactive states, respectively. The following previously published datasets were used: EMD-20849, Cryo-EM structure of type 3 IP3 receptor revealing presence of a self-binding peptide[10]. 6UQK, Cryo-EM structure of type 3 IP3 receptor revealing presence of a self-binding peptide[10]. 6DRC, High IP3 $Ca^{2+}$ human type 3 1,4,5-inositol trisphosphate receptor[6]. 6DQV Class 2 IP3-bound human type 3 1,4,5-inositol trisphosphate receptor[6]. 5TAL, Structure of rabbit RyR1 (Caffeine/ ATP/$Ca^{2+}$ dataset, class 1 and 2)[25]. 7M6A, High resolution structure of the membrane embedded skeletal muscle ryanodine receptor[29]. Reagents and other materials will be available upon request from E.K. with a completed materials transfer agreement.

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

## Acknowledgements

We thank Dr. Kunpeng Lee for cryo-EM data collection at Case Western Reserve University. We thank Theo Humphries and other support staff at the Pacific Northwest Center for Cryo-EM (PNCC), Drs. Elad Binshtein, Melissa Chambers, and Scott Collier at the Cryo-EM facility at Vanderbilt University for their assistance with cryo-EM sample screening. We thank Drs. Hassane Mchaourab, Terunaga Nakagawa, and Silvia Ravera for discussions and review of the manuscript. This work was conducted in part using the CPU and GPU resources of the Advanced Computing Center for Research and Education (ACCRE) at Vanderbilt University. We used the DORS storage system supported by the U.S. National Institute of Health (NIH) (S10RR031634 to Jarrod Smith). This work was supported by the NIH (R01GM141251 to E.K.), Vanderbilt University, Vanderbilt

Diabetes and Research Training Center (NIH P30DK020593 to E.K.), and the Molecular Biophysics Training Program (NIH T32GM008320 to E.A.S.).

## Author contributions

E.K. conceived the project and performed cryo-EM data analysis; E.A.S. optimized and performed protein expression and purification; H.T. performed grid preparation for cryo-EM. All authors contributed to the preparation of the manuscript.

## Competing interests

The authors declare no competing interests.
