## [Peer Review File · Nature Communications]

Structural basis for activation and gating of IP3 receptorsReviewers' Comments:

Reviewer #1:

Remarks to the Author:

The manuscript by Schmitz et.al. reports structures of human inositol triphosphate receptor (hIP3R) obtained by single particle cryo-EM in the presence of IP3, ATP and nanomolar concentration of free calcium. Using 3D classification authors were able to resolve several conformations of hIP3R including open ion channel conformation that so far remained elusive. The ATP binding site was resolved for the first time as well.

The manuscript provides new insight into structural states associated with function of IP3R which will have a significant impact on the field.

A number of questions need to be addressed before the manuscript can be published.

- 1) Authors describe an inactive state for which density for a part of the cytoplasmic domain is missing in the 3D reconstruction. It is not clear at all whether such an assignment makes sense. During protein purification and preparation of cryo-EM samples, some particles may degrade (partially or completely unfold) potentially resulting in reconstruction interpreted as inactive state. Unless additional arguments can be provided supporting assignment of this class to a physiologically relevant state, this conformation should be considered as a degraded protein and be removed from the main text and Figure 1. It should be checked if it displays any preferred particle orientation as well. Moreover, such conformation while being reconstructed from a large number of particles and therefore potentially prevalent, was not observed by other groups under very similar buffer conditions.
- 2) The abstract is confusing in that it suggests that multiple datasets were collected under different conditions while authors mean that various conformations were resolved from a single biochemically defined state and conclusions regarding bound calcium, for example, come from interpretation of the density map. This needs to be rephrased.
- 3) The density in Figure 1 does not allow us to judge whether trans-membrane helices are well resolved. The images need to be improved.
- 4) From Extended Table 1 it is apparent that the open or active state represents only around 4% of all particles for which high-resolution reconstruction was obtained. This fact is not mentioned in the manuscript. It should be explicitly mentioned and discussed.
- 5) In the section 'Priming of hIP3R-3 for activation', authors should describe conformational changes in the context of known IP3R structures obtained with activating ligands references 6 and 8 otherwise it sounds as if structures in presence of activators have never been described previously.
- 6) The manuscript describes pre-active states A, B and C as being Ca²⁺ free states. At the same time figure 1b does show density in the area where Ca²⁺ is bound in active and inactive states. How was the presence of bound calcium assessed? This needs to be explained clearly because it is one of the critical points in the interpretation of the structural data.
- 7) It would be very valuable for readers to see a discussion at the end of the manuscript describing reasons why other attempts to resolve open state failed and what is (if any) functional significance of the so small occupancy of the active state in spite of the all the efforts made to prepare the channel in active state? Can the fraction of active state be further improved in the future?

Reviewer #2:

Remarks to the Author:

In this manuscript, the authors investigate various conformations of IP3 Receptor isoform 3. By introducing a cocktail of ATP, calcium, and IP3, the authors collected cryo-EM data that led to 5 different structures, which seem to represent different snapshots of an IP3R that becomes gradually more active. The study is the first to report a bound ATP to an IP3R, and the binding site seems similar (although not identical) to the one found in RyRs.

Overall, this study is of relevance, as it describes novel conformations of the IP3R. I do have some issues with the quality of the maps, which I can think can be improved via additional processing, as explained below. I also have some strong suggestions around interpretation and comparisons of the structures.

Major comments:

- Although the maps look ok, their quality could be improved via several procedures. This includes density modification (e.g. as implemented in Phenix) or local masking. The latter can also be very powerful in combination with symmetry expansion. I suggest the authors to perform density modification, but separately also to perform local masking/refinement (with and without symmetry expansion) around the regions where the ligands are bound. This may greatly improve the local density and hence chemical interpretation of the binding sites.

The above is critical for ATP, because there is still quite some ambiguity on the conformation of ATP, as it has been modeled in. In the case of the 'open' structure, for example, the gamma phosphate is curved back to the first oxygen of the triphosphate, which is energetically unlikely. The binding site is similar to what has been observed in Ryanodine Receptors (RyRs), which makes sense given the close evolutionary relationship, but the exact conformation of ATP differs substantially, with the adenine ring oriented in the opposite way (e.g. look at PDB 7M6A or 7M6L). Whereas this does not disprove that the author's binding mode is wrong, it does warrant further investigation through: a) producing the best possible locally refined map and b) build in ATP in different conformations and calculate map correlation coefficients for the various modes. This will take some time but this reviewer deems this as very important.

Also, the structure referred to as 'inactive' has a density that is insufficient to observe the presence or absence of the 'inhibitory' calcium. The authors should definitely try symmetry expansion + local masking here to try and improve the density.

- Further to the ATP binding, it would be good for the authors to show a detailed comparison of the ATP binding site in RyRs and IP3Rs. Extended figure 5 shows a superposition in cartoon model, but no details are shown. Which exact residues are conserved? Which ones are different? The different binding mode of ATP in both structures is surprising. Other than trying to improve the local density as explained above, it would be good, if confirmed, to explain 'why' this is the case, e.g. by pointing out more detailed comparisons. E.g. are there steric clashes taking the conformation of ATP from one structure and putting it in the other one (through superposition). Are there additional interactions in one channel that do not occur in the other? As multiple RyR structures have been reported with ATP bound, the authors may want to pick representatives from different publications.

- As the authors have now a nice ensemble of conformational states available, it would be good to see how known sequence variants/disease mutations affect this process. Hundreds of sequence variants can be found in databases (e.g. a few variants in ITPR3 are reported on clinvar, but many more are available for ITPR1), often found in patients with various disorders. The impact of this manuscript would rise substantially if these were mapped on the structures and analyzed in light of the conformational changes. Which ones can be mapped to the mobile interfaces? Which ones can be mapped to binding sites for the small molecules described in this study? Do some disease phenotypes correlate with the location? Given the many variants found in IP3R1, some generalizations may be possible.

Minor comments:

- Validation reports: please update these to the final validation reports. The first page of each clearly states "do not submit to journals". Also, please upload the half maps so the PDB validation can generate FSC curves.
- Video: this is very useful. I do recommend to build in a short 1 or 2-second pause at each intermediate step. Right now the video is very fast and continuous, so it's hard to gauge what step we're looking at.
- There is a discrepancy in the conditions as reported in the results and in the Methods. E.g. p10 does not mention the presence of EDTA in the purified sample. Please address this.

Reviewer #3:

Remarks to the Author:

The IP3 receptor plays a fundamental role in calcium signaling. Still, molecular-level knowledge on its function and regulation is very much in its infancy. This study on the activation mechanism is therefore very significant, providing the first view of what very much appears to be an active and open state. This allows the observation for the first time of a path from primed to active to putative desensitized state. This is a significant advance towards understanding the function of this channel and understanding the interplay between IP3 and calcium during activation.

Overall, the work appropriately supports the conclusions made and the figures appropriately represent the features of the EM maps. The most important point that appears valid upon close inspection of the maps and models is the absence of calcium in the calcium binding site at the interface of the JD and ARM3 domains in the states called pre-active.

The work meets the standards of the field in terms of methodology and data quality, but some methodologies have been omitted that could have greatly improved the quality of the maps and allowed more definitive statements in the manuscript. It does feel like a missed opportunity. These channels are very large and flexible, which explains the limited resolution of the study despite the large number of micrographs collected. Focused classification on the core of the channel allowed the separation of fine differences in states. Similar methods, such as multi-body refinement, should have been used for refinement to improve the resolution of the maps in all portions of the molecule. This would have allowed to obtain much better resolutions throughout and would have resolved the state of the second calcium binding site, which is a significant caveat in the story. Inspection of the maps strongly suggests that the calcium is indeed bound in this second binding site, but better resolution would settle the debate. While not absolutely necessary for publication, it would significantly improve the manuscript and its impact.

If resolution cannot be improved in this region in the context of a revision, a figure describing the similarity in conformation of this region compared to the high-calcium structure should be provided to support the fact that calcium is likely to be bound there.

Another missed opportunity is the lack of proper description of the rejected classes. This is akin to discarding data points on a graph. A more thorough description of those classes should be provided given that they are at resolutions sufficient to assess their conformational states. This would better support the author's sequential activation mechanism model. While pre -> active state is logical, there is currently no support for the pre-A->pre-B->pre-C sequence of events.

More minor points:

"ATP binds to a similar location near the zinc finger motif in RyRs20, but residues forming the binding pocket are distinct "

This statement needs to be more specific. How distinct?

"However, the CTD is poorly resolved in hIP3R-3 structures, including the active state, indicating high flexibility of the CTD relative to the rest of the protein and suggesting that the structural rearrangements of this domain are unlikely to create enough force to open the channel "

Conformational changes in this region should be better assessed using modeling and careful map comparisons at lower resolutions. Local filtering of low-resolution regions or global filtering of the maps to assess conformational changes in these regions should be done. In particular, the loose/inactive conformations should be displayed with the cytoplasmic regions filtered at a lower resolution so that they are visible. The current display of those maps misrepresents them as partially disordered when they are not.

In methods,

"Purified hIP3R-3 was supplemented with 500 μ M IP3 (from 10 mM stock in water), 0.1 mM CaCl₂, and 5 mM ATP (from 100 mM stock, pH 7.2). 2.0 μ L of the protein sample was applied to 300 mesh Cu Quantifoil 1.2/1.3 grids (Quantifoil Microtools) that were glow discharged for 20 seconds at 25 mA. "

Was edta or egta supplied in the buffer?

Was whatman paper treated w EGTA or ashless?

How was the free ca²⁺ concentration in solution assessed?

RESPONSE TO REVIEWERS:

We thank the reviewers for critically evaluating our recent manuscript. We are delighted by the generally positive comments that our manuscript received. We are also grateful for the constructive and detailed comments that helped us improve our manuscript. We have updated the manuscript and figures to reflect all of the changes requested.

Please find our point by point response (colored red) to the reviewers' comments directly underneath each comment below. The changes in our manuscript can also be tracked in our resubmitted document.

Reviewer #1 (Remarks to the Author):

The manuscript by Schmitz et.al. reports structures of human inositol triphosphate receptor (hIP3R) obtained by single particle cryo-EM in the presence of IP3, ATP and nanomolar concentration of free calcium. Using 3D classification authors were able to resolve several conformations of hIP3R including open ion channel conformation that so far remained elusive. The ATP binding site was resolved for the first time as well. The manuscript provides new insight into structural states associated with function of IP3R which will have a significant impact on the field.

A number of questions need to be addressed before the manuscript can be published.

1) Authors describe an inactive state for which density for a part of the cytoplasmic domain is missing in the 3D reconstruction. It is not clear at all whether such an assignment makes sense. During protein purification and preparation of cryo-EM samples, some particles may degrade (partially or completely unfold) potentially resulting in reconstruction interpreted as inactive state. Unless additional arguments can be provided supporting assignment of this class to a physiologically relevant state, this conformation should be considered as a degraded protein and be removed from the main text and Figure 1. It should be checked if it displays any preferred particle orientation as well. Moreover, such conformation while being reconstructed from a large number of particles and therefore potentially prevalent, was not observed by other groups under very similar buffer conditions.

In the revised manuscript, we performed local refinement by masking different regions of the particles in response to reviewers 2 and 3's comments. The resulting maps (Supplementary Fig. 6 in the revised manuscript) revealed better-resolved maps for the cytoplasmic domains of the inactive structure, indicating that the missing parts of the cryo-EM map are due to the higher flexibility of the region relative to the rest of the protein rather than being degraded or unfolded. Using the new maps, we built a new model that comprised most of the receptor and compared it to the IP₃R-3 structure obtained at high Ca²⁺ concentrations (Supplementary Fig. 8 in the revised manuscript) (Paknejad&Hite, 2018). The comparison reveals that the structures are highly similar, especially around the ARM2 domain that moves in the Ca²⁺ inhibited conformation, validating our initial assessment of this class as inactive. We modified the text to clarify our reasoning in this class assignment as inactive, as follows;

Line 100 on page 5: “.....In the fifth structure, the channel is closed, the activatory Ca^{2+} binding site is occupied, and the intersubunit interactions of the cytoplasmic domains are lost (Fig. 1d; Supplementary Fig. 6). The structure is highly similar to the hIP₃R-3 structures obtained in the presence of inhibitory Ca^{2+} concentrations, except for βTf1 , which moves closer to the ARM1 (Supplementary Fig. 8)⁶. Most notably, ARM2 adopted the same conformation relative to ARM1 and CLD, creating the binding site for the second Ca^{2+} observed at high Ca^{2+} concentrations (Supplementary Fig. 8). While these similarities suggest a Ca^{2+} ion occupies this site and the structure represents the Ca^{2+} inhibited state, the quality of the map around the region did not allow accurate inspection of the presence of Ca^{2+} . Therefore, we refer to the structure as “inactive” while it remains unclear if it represents a desensitized state that hIP₃R-3 adopts without additional Ca^{2+} binding or an inhibited state forced by binding of additional Ca^{2+} to an inhibitory site.”

2) The abstract is confusing in that it suggests that multiple datasets were collected under different conditions while authors mean that various conformations were resolved from a single biochemically defined state and conclusions regarding bound calcium, for example, come from interpretation of the density map. This needs to be rephrased.

We modified the abstract to clarify that all conformations described in the manuscript were obtained from a single dataset.

3) The density in Figure 1 does not allow us to judge whether trans-membrane helices are well resolved. The images need to be improved.

We remade Figure 1 using the composite maps created by combining maps obtained after local refinements. The new figure shows better-resolved maps for the entire protein, including the transmembrane helices.

4) From Extended Table 1 it is apparent that the open or active state represents only around 4% of all particles for which high-resolution reconstruction was obtained. This fact is not mentioned in the manuscript. It should be explicitly mentioned and discussed.

We added a new paragraph at the end of the section, Cryo-EM structures of hIP₃R-3 gating conformations, mentioning the low number of particles in the active state compared to the pre-active states. The paragraph reads as;

Line 112 on page 5: “It is important to note that our initial 3D classification runs resulted in two major classes grouping the pre-active and active structures into one class and the inactive structure into another (Supplementary Fig. 1). It was essential to perform another round of 3D classification focusing only on the core of the protein to separate the particles in the active state from the pre-active states, potentially due to subtle differences in the overall structures and the much fewer number of particles in the active state (20,039 particles compared to 346,684 particles in the pre-active states) (Supplementary Fig. 1; Supplementary Table 1).”

5) In the section ‘Priming of hIP₃R-3 for activation’, authors should describe conformational changes in the context of known IP₃R structures obtained with activating ligands references 6 and 8 otherwise it sounds as if structures in presence of activators have never been described previously.

We changed the beginning of this section, as stated below, to clarify that IP₃ induced changes we observed in pre-active A structure are identical or highly similar to the changes described in previous studies, involving both cryo-EM and X-ray crystallography, with additional citations. To our knowledge, the conformational changes observed in pre-active B and pre-active C states are observed for the first time in our study. The sentence now reads as follows.

Line 122 on page 6: “...The pre-active A structure is almost identical to the previously published IP₃-bound hIP₃R-3 structure⁶ and reveals that IP₃ binding causes the ARM1 to rotate about 23° relative to the βTF-2, causing global conformational changes within the cytoplasmic domains, as observed in previous cryo-EM and X-ray crystallography experiments^{6,8,19-22} (Supplementary Fig. 9).....”

6) The manuscript describes pre-active states A, B and C as being Ca²⁺ free states. At the same time figure 1b does show density in the area where Ca²⁺ is bound in active and inactive states. How was the presence of bound calcium assessed? This need to be explained clearly because it is one of the critical points in the interpretation of the structural data.

The presence of Ca²⁺ was assessed by visual inspection of the non-protein density in the cryo-EM maps. The density seen in Fig. 1b is of the protein residues. The new figure prepared using the maps obtained after local refinement allows a better comparison of the Ca²⁺ binding sites.

7) It would be very valuable for readers to see a discussion at the end of the manuscript describing reasons why other attempts to resolve open state failed and what is (if any) functional significance of the so small occupancy of the active state in spite of the all the efforts made to prepare the channel in active state? Can the fraction of active state be further improved in the future?

We thank the reviewer for this great suggestion. We added two paragraphs to the beginning and end of the section called “Cryo-EM structures of hIP₃R-3 gating conformations”, rather than having a discussion at the end of the manuscript. These paragraphs describe the challenges in obtaining the active state structure and our approaches to overcoming them, which we believe were critical for our success.

Line 70 on page 5: “Bimodal regulation of IP₃R activity by Ca²⁺ complicates sample preparation because of the requirement for fine adjustment of Ca²⁺ concentration to trap the channel in the open conformation. Although free Ca²⁺ concentrations in solutions can easily be controlled by using Ca²⁺ buffers such as EGTA or BAPTA, it becomes challenging during sample preparation for cryo-EM due to the small volumes used. Typically, a 2-3 μl protein sample is applied to a cryo-grid, but more than 99% of the sample volume is lost during grid preparation due to extensive blotting with filter paper¹⁸. During this time, the samples contact filter paper and cryo-grids, containing various amounts of Ca²⁺. Small sample volumes and short time frames may reduce these buffers' efficiency, causing the free Ca²⁺ concentration to increase to inhibitory levels prior to sample freezing. In order to maximize the chances of obtaining particles in the active state, we prepared the sample in: 1) EDTA, which has ~200 fold faster binding kinetics to Ca²⁺ than EGTA¹⁹, a common Ca²⁺ chelator, and is more likely to chelate excess Ca²⁺ and other divalent cations within the short period prior to the sample plunging, 2) ATP, which increases the open probability of IP₃Rs and dampens the inhibitory effect of Ca²⁺^{16,17}, and 3) high concentrations of IP₃”.

Line 112 on page 5: “It is important to note that our initial 3D classification runs resulted in two major classes grouping the pre-active and active structures into one class and the inactive structure into another (Supplementary Fig. 1). It was essential to perform another round of 3D classification focusing only on the core of the protein to separate the particles in the active state from the pre-active states, potentially due to subtle differences in the overall structures and the much fewer number of particles in the active state (20,039 particles compared to 346,684 particles in the pre-active states) (Supplementary Fig. 1; Supplementary Table 1).”

Reviewer #2 (Remarks to the Author):

In this manuscript, the authors investigate various conformations of IP3 Receptor isoform 3. By introducing a cocktail of ATP, calcium, and IP3, the authors collected cryo-EM data that led to 5 different structures, which seem to represent different snapshots of an IP3R that becomes gradually more active. The study is the first to report a bound ATP to an IP3R, and the binding site seems similar (although not identical) to the one found in RyRs.

Overall, this study is of relevance, as it describes novel conformations of the IP3R. I do have some issues with the quality of the maps, which I can think can be improved via additional processing, as explained below. I also have some strong suggestions around interpretation and comparisons of the structures.

Major comments:

- Although the maps look ok, their quality could be improved via several procedures. This includes density modification (e.g. as implemented in Phenix) or local masking. The latter can also be very powerful in combination with symmetry expansion. I suggest the authors to perform density modification, but separately also to perform local masking/refinement (with and without symmetry expansion) around the regions where the ligands are bound. This may greatly improve the local density and hence chemical interpretation of the binding sites.

We thank the reviewer for this great suggestion, which helped us improve the quality of the maps considerably. We performed both density modification and local refinement for all the structures described in the manuscript. While we did not observe significant improvement on density modified maps, the local refinement helped us resolve a large portion of the poorly defined regions in the original maps and improve the overall quality of the maps around the regions where ligands are bound. We remodeled the structures into the composites maps created from these locally refined maps and updated the figures. Supplementary Figs. 2-6 summarize the protocol we performed and show the new, improved maps.

The above is critical for ATP, because there is still quite some ambiguity on the conformation of ATP, as it has been modeled in. In the case of the ‘open’ structure, for example, the gamma phosphate is curved back to the first oxygen of the triphosphate, which is energetically unlikely. The binding site is similar to what has been observed in Ryanodine Receptors (RyRs), which makes sense given the close evolutionary relationship, but the exact conformation of ATP differs substantially, with the adenine ring oriented in the opposite way (e.g. look at PDB 7M6A or 7M6L). Whereas this does not disprove that the author’s binding mode is wrong, it does

warrant further investigation through: a) producing the best possible locally refined map and b) build in ATP in different conformations and calculate map correlation coefficients for the various modes. This will take some time but this reviewer deems this as very important.

As described in response to the previous comment, the local refinement approach improved the maps around the ATP binding site in all states, and the resulting maps, as shown in Fig. 4a in the revised manuscript, clearly reveal the orientation of the adenine ring and position of the phosphate moieties.

Also, the structure referred to as 'inactive' has a density that is insufficient to observe the presence or absence of the 'inhibitory' calcium. The authors should definitely try symmetry expansion + local masking here to try and improve the density.

We performed local refinements with and without symmetry expansion to improve the map quality around the Ca^{2+} binding site. We also used masks covering different regions around the site and applied different search parameters. We were able to improve the overall map quality for the NTD that harbors the inhibitory Ca^{2+} binding site. Unfortunately, we did not obtain any maps useful for unambiguous determination of the presence of Ca^{2+} at this site. We used the improved maps to build a better model for the NTD and compared this model to the IP₃R-3 structure obtained at high Ca^{2+} concentrations. In the revised manuscript, we prepared a new figure (Supplementary Fig. 8) showing the similarities between the structures, especially at the inhibitory Ca^{2+} binding site, and changed the description of the inactive state as shown below to emphasize these similarities.

Line 100 on page 5: ".....In the fifth structure, the channel is closed, the activatory Ca^{2+} binding site is occupied, and the intersubunit interactions of the cytoplasmic domains are lost (Fig. 1d; Supplementary Fig. 6). The structure is highly similar to the hIP₃R-3 structures obtained in the presence of inhibitory Ca^{2+} concentrations, except for βTf1 , which moves closer to the ARM1 (Supplementary Fig. 8)⁶. Most notably, ARM2 adopted the same conformation relative to ARM2 and CLD, creating the binding site for the second Ca^{2+} observed at high Ca^{2+} concentrations (Supplementary Fig. 8). While these similarities suggest a Ca^{2+} ion occupies this site and the structure represents the Ca^{2+} inhibited state, the quality of the map around the region did not allow accurate inspection of the presence of Ca^{2+} . Therefore, we refer to the structure as "inactive", while it remains unclear if it represents a desensitized state that hIP₃R-3 adopts without additional Ca^{2+} binding or an inhibited state forced by binding of additional Ca^{2+} to an inhibitory site."

- Further to the ATP binding, it would be good for the authors to show a detailed comparison of the ATP binding site in RyRs and IP3Rs. Extended figure 5 shows a superposition in cartoon model, but no details are shown. Which exact residues are conserved? Which ones are different? The different binding mode of ATP in both structures is surprising. Other than trying to improve the local density as explained above, it would be good, if confirmed, to explain 'why' this is the case, e.g. by pointing out more detailed comparisons. E.g. are there steric clashes taking the conformation of ATP from one structure and putting it in the other one (through superposition). Are there additional interactions in one channel that do not occur in the other? As multiple RyR structures have been reported with ATP bound, the authors may want to pick representatives from different publications.

We changed the figure that compares the structures of the ATP binding sites of IP₃R-3 and RyR-1 (PDB IDs: 5TAL and 7M6A) to show the key residues that are important for ATP binding and differ among these receptors (Supplementary Fig. 10c,d in the revised manuscript). We discussed these differences in the revised manuscript as follow;

Line 167 on page 8: “ATP binds to a similar location near the zinc finger motif in RyRs^{25,29,30}. However, its binding mode differs, potentially due to the differences in the residues that form the binding pocket, most notably the basic residues interacting with the phosphate moieties (Supplementary Fig. 10a, c, d). In RyR-1s, the phosphate moieties interact with K4211, K4214, and R4215, all located on a single helix (Supplementary Fig. 10 c, d). In hIP₃R-3, there is only a single lysine residue (K2152) on the corresponding helix, and the phosphate moieties interact with K2560, located on the opposite side of the binding pocket. A leucine residue (L4980) occupies this position in RyR-1s. The differences in the number and location of the basic residues likely force the phosphate moieties of ATP to adopt different conformations. Furthermore, F2156 in hIP₃R-3 points toward the adenosine binding pocket, prohibiting ATP from adopting the conformations observed in RyR-1s due to steric clash in hIP₃R-3s (Supplementary Fig. 10 c, d).”

- As the authors have now a nice ensemble of conformational states available, it would be good to see how known sequence variants/disease mutations affect this process. Hundreds of sequence variants can be found in databases (e.g. a few variants in ITPR3 are reported on ClinVar, but many more are available for ITPR1), often found in patients with various disorders. The impact of this manuscript would rise substantially if these were mapped on the structures and analyzed in light of the conformational changes. Which ones can be mapped to the mobile interfaces? Which ones can be mapped to binding sites for the small molecules described in this study? Do some disease phenotypes correlate with the location? Given the many variants found in IP3R1, some generalizations may be possible.

We mapped the mutations associated with diseases or annotated as pathogenic or likely pathogenic in ClinVar database for the three IP₃R subtypes on the hIP₃R-3 structure and prepared a new figure showing the position of these residues on the receptor (Supplementary Fig. 12a). In addition to the previously identified hot spot around the IP₃ binding site, we observed clustering of the mutations around the constriction site of the pore at the TMD. We prepared a figure with the close up view of this site (Supplementary Fig. 12b) and discussed the potential effect of the mutations on channel activity in the following paragraph.

Line 208 on page 9: “Several missense mutations identified in the IP₃R subtypes are associated with diseases, including spinocereblar ataxia, Gillespie syndrome, anhidrosis, and neck squamous cell carcinoma (Supplementary Fig. 12; reviewed in ^{32,35,36}). Perhaps not surprisingly, most of these mutations are localized around the IP₃ binding site and alter IP₃ binding affinity^{32,35-37}. Another hot spot for these mutations is the constriction site of the pore, which undergoes conformational changes during channel opening (Supplementary Fig. 12). It is plausible that these mutations either affect the Ca²⁺ permeability (e.g., mutation of N2510³⁸ or I2517³⁹) or restrict conformational changes required for dilation of the pore (e.g., mutation of G2514³¹). Two of the mutated residues (T2519⁴⁰ and F2520⁴¹) interact with the residues on the S4-S5 linker, which couples the tilting of the pVSD to the bulging of the constriction site (Supplementary Fig. 12b). Mutations of these residues are likely to impair this coupling and thus hinder gating.”

Minor comments:

- Validation reports: please update these to the final validation reports. The first page of each clearly states “do not submit to journals”. Also, please upload the half maps so the PDB validation can generate FSC curves.

We updated the final versions of the validation reports of the submitted PDB coordinates and the EM maps. We uploaded half maps, unsharpened and composite maps to the database as well.

- Video: this is very useful. I do recommend to build in a short 1 or 2-second pause at each intermediate step. Right now the video is very fast and continuous, so it’s hard to gauge what step we’re looking at.

We remade the video using the updated coordinates and incorporated the changes suggested.

- There is a discrepancy in the conditions as reported in the results and in the Methods. E.g. p10 does not mention the presence of EDTA in the purified sample. Please address this.

EDTA was present in the final SEC buffer. To clarify this discrepancy, we made the following changes to the text;

Line 85 on page 4: “The final hIP₃R-3 sample was purified in the presence of 1 mM EDTA and supplemented with 0.5 mM IP₃, 5 mM ATP, and 0.1 mM CaCl₂ before preparing cryo-grids.....”

Line 295 on page 13: “Purified hIP₃R-3 in the SEC buffer containing 1 mM EDTA was supplemented with 500 μM IP₃ (from 10 mM stock in water), 0.1 mM CaCl₂, and 5 mM ATP (from 100 mM stock, pH 7.2). 2.0 μL of the protein sample was applied to 300 mesh Cu Quantifoil 1.2/1.3 grids (Quantifoil Microtools) that were glow discharged for 20 seconds at 25 mA....”

Reviewer #3 (Remarks to the Author):

The IP₃ receptor plays a fundamental role in calcium signaling. Still, molecular-level knowledge on its function and regulation is very much in its infancy. This study on the activation mechanism is therefore very significant, providing the first view of what very much appears to be an active and open state. This allows the observation for the first time of a path from primed to active to putative desensitized state. This is a significant advance towards understanding the function of this channel and understanding the interplay between IP₃ and calcium during activation.

Overall, the work appropriately supports the conclusions made and the figures appropriately represent the features of the EM maps. The most important point that appears valid upon close inspection of the maps and models is the absence of calcium in the calcium binding site at the interface of the JD and ARM3 domains in the states called pre-active.

The work meets the standards of the field in terms of methodology and data quality, but some

methodologies have been omitted that could have greatly improved the quality of the maps and allowed more definitive statements in the manuscript. It does feel like a missed opportunity. These channels are very large and flexible, which explains the limited resolution of the study despite the large number of micrographs collected. Focused classification on the core of the channel allowed the separation of fine differences in states. Similar methods, such as multi-body refinement, should have been used for refinement to improve the resolution of the maps in all portions of the molecule. This would have allowed to obtain much better resolutions throughout and would have resolved the state of the second calcium binding site, which is a significant caveat in the story. Inspection of the maps strongly suggests that the calcium is indeed bound in this second binding site, but better resolution would settle the debate. While not absolutely necessary for publication, it would significantly improve the manuscript and its impact.

We thank the reviewer for the great suggestions, which helped us improve the quality of the maps considerably. As discussed in response to reviewer 2, we performed both density modification and local refinement for all the structures described in the manuscript. While we did not observe significant improvement on density modified maps, the local refinement helped us resolve a large portion of the poorly defined regions in the original maps and improve the overall quality of the maps around the regions where ligands are bound. We remodeled the structures into the composites maps created from these locally refined maps and updated the figures. Supplementary Figs. 2-6 summarize the protocol we performed and show the new, improved maps.

We performed local refinements with and without symmetry expansion for the structure in the inactive state to improve the map quality around the Ca^{2+} binding site. We also used masks covering different regions around the site and applied different search parameters. Although the overall map quality for the NTD that harbors the inhibitory Ca^{2+} binding site improved significantly, we did not obtain any maps useful for unambiguous determination of the presence of Ca^{2+} at this site.

If resolution cannot be improved in this region in the context of a revision, a figure describing the similarity in conformation of this region compared to the high-calcium structure should be provided to support the fact that calcium is likely to be bound there.

We used the improved maps to build a better model for the NTD and compared this model to the $\text{IP}_3\text{R-3}$ structure obtained at high Ca^{2+} concentrations. In the revised manuscript, we prepared a new figure (Supplementary Fig. 8) showing the similarities between the structures, especially at the inhibitory Ca^{2+} binding site, and changed the description of the inactive state as shown below to emphasize these similarities.

Line 100 on page 5: "...In the fifth structure, the channel is closed, the activatory Ca^{2+} binding site is occupied, and the intersubunit interactions of the cytoplasmic domains are lost (Fig. 1d; Supplementary Fig. 6). The structure is highly similar to the $\text{hIP}_3\text{R-3}$ structures obtained in the presence of inhibitory Ca^{2+} concentrations, except for βTF1 , which moves closer to the ARM1 (Supplementary Fig. 8)⁶. Most notably, ARM2 adopted the same conformation relative to ARM2 and CLD, creating the binding site for the second Ca^{2+} observed at high Ca^{2+} concentrations (Supplementary Fig. 8). While these similarities suggest a Ca^{2+} ion occupies this site and the structure represents the Ca^{2+} inhibited state, the quality of the map around the region did not allow accurate inspection of the presence of Ca^{2+} . Therefore, we refer to the structure as

“inactive” while it remains unclear if it represents a desensitized state that hIP₃R-3 adopts without additional Ca²⁺ binding or an inhibited state forced by binding of additional Ca²⁺ to an inhibitory site.”

Another missed opportunity is the lack of proper description of the rejected classes. This is akin to discarding data points on a graph. A more thorough description of those classes should be provided given that they are at resolutions sufficient to assess their conformational states. This would better support the author's sequential activation mechanism model. While pre → active state is logical, there is currently no support for the pre-A → pre-B → pre-C sequence of events.

After focused 3D classification, we performed non-uniform refinement for each class before rejecting any of the classes. Two of the rejected classes were almost identical to the classes designated as pre-active A and pre-active B, but the map quality was not as good. During the reevaluation of these classes for the resubmission, we observed that the map quality was slightly better when we combined the pre-active B class and the rejected class with a similar structure. For the pre-active A, combining the two classes lowered the map quality. Therefore, we kept them separate but indicated that the rejected class was similar to pre-active A in Supplementary Fig. 1. The rejected class with ~19K particles, poorly resolved TMD, especially the selectivity filter and S6 helix. It was difficult and unreliable to make a comparison with the other classes. The map quality for the last rejected class was very poor, and we did not attempt to place models in.

More minor points:

"ATP binds to a similar location near the zinc finger motif in RyRs20, but residues forming the binding pocket are distinct "

This statement needs to be more specific. How distinct?

To clarify this statement and provide more details, we changed the figure that compares the structures of the ATP binding sites of IP₃R-3 and RyR-1 to show the key residues that are important for ATP binding and differ among these receptors (Supplementary Fig. 10c,d in the revised manuscript). We discussed these differences in the revised manuscript as follow;

Line 167 on page 8: “ATP binds to a similar location near the zinc finger motif in RyRs^{25,29,30}. However, its binding mode differs, potentially due to the differences in the residues that form the binding pocket, most notably the basic residues interacting with the phosphate moieties (Supplementary Fig. 10a, c, d). In RyR-1s, the phosphate moieties interact with K4211, K4214, and R4215, all located on a single helix (Supplementary Fig. 10 c, d). In hIP₃R-3, there is only a single lysine residue (K2152) on the corresponding helix, and the phosphate moieties interact with K2560, located on the opposite side of the binding pocket. A leucine residue (L4980) occupies this position in RyR-1s. The differences in the number and location of the basic residues likely force the phosphate moieties of ATP to adopt different conformations. Furthermore, F2156 in hIP₃R-3 points toward the adenosine binding pocket, prohibiting ATP from adopting the conformations observed in RyR-1s due to steric clash in hIP₃R-3s (Supplementary Fig. 10 c, d).”

"However, the CTD is poorly resolved in hIP3R-3 structures, including the active state, indicating high flexibility of the CTD relative to the rest of the protein and suggesting that the

structural rearrangements of this domain are unlikely to create enough force to open the channel "

Conformational changes in this region should be better assessed using modeling and careful map comparisons at lower resolutions. Local filtering of low-resolution regions or global filtering of the maps to assess conformational changes in these regions should be done.

In the revised manuscript, we used unsharpened maps at two different threshold values to compare the maps around the CTD (Supplementary Fig. 12). We modeled unregistered poly-alanine peptides for the relatively better defined coiled-coiled region and used the new models to compare the conformational changes at the CTD. The coiled-coil region remains unchanged in the pre-active states but rotates about 20° and moves closer to the TMD by ~6 Å in the active state. These conformational changes are shown in Supplementary Fig. 13 and discussed in the revised manuscript.

Line 220 on page 10: "The CTD, extending from the JD along the symmetry axis, forms a left-handed coiled-coil motif before interacting with the βTF2 of the neighboring subunit. The density for the CTD was poorly resolved in all of the states (Supplementary Fig. 13a, b). However, the coiled-coil motifs were visible in the unsharpened maps in the pre-active and active states, enabling us to model poly-alanine peptides without assigned registries (Supplementary Fig. 13a, b). The densities for the extensions from the coiled-coil motif towards the βTF2 become visible when viewed at lower thresholds, whereas the linkers between the JD and the coiled-coil motif remain invisible, indicating higher flexibility for this region (Supplementary Fig. 13a, b). We did not observe any interpretable density for the CTD in the inactive state (Supplementary Fig. 13a, b).

For IP₃R-1, the CTD was proposed to transmit the conformational changes induced by IP₃ at the NTD to the JD⁸. In IP₃R-3, there are no apparent changes on the coiled-coil motif in the pre-active states, but the coiled-coil motif rotates about 20° around the symmetry axis and moves closer to the TMD by 6 Å in the active state (Supplementary Fig. 13c, d). However, the linker between the coiled-coil motif and JD remains flexible, suggesting that the structural rearrangements of this domain are not directly enforcing the channel opening (Supplementary Fig. 13). In line with these observations, removing CTD residues interacting with the βTF2 or swapping the C-terminal region of IP₃R-1 with the RyRs, which lack the extended CTD, did not diminish receptor activation^{21,24,42}."

In particular, the loose/inactive conformations should be displayed with the cytoplasmic regions filtered at a lower resolution so that they are visible. The current display of those maps misrepresents them as partially disordered when they are not.

The local refinements improved the map quality around the cytoplasmic region of the inactive conformation, and we used the composite map generated from these maps in the figures.

In methods,

"Purified hIP3R-3 was supplemented with 500 μM IP3 (from 10 mM stock in water), 0.1 mM CaCl₂, and 5 mM ATP (from 100 mM stock, pH 7.2). 2.0 μL of the protein sample was applied to 300 mesh Cu Quantifoil 1.2/1.3 grids (Quantifoil Microtools) that were glow discharged for 20 seconds at 25 mA. "

- Was edta or egta supplied in the buffer?

EDTA was present in the final SEC buffer. To clarify this discrepancy, we made the following changes to the text;

Line 85 on page 4: “The final hIP₃R-3 sample was purified in the presence of 1 mM EDTA and supplemented with 0.5 mM IP₃, 5 mM ATP, and 0.1 mM CaCl₂ before preparing cryo-grids.....”

Line 295 on page 13: “Purified hIP₃R-3 in the SEC buffer containing 1 mM EDTA was supplemented with 500 μM IP₃ (from 10 mM stock in water), 0.1 mM CaCl₂, and 5 mM ATP (from 100 mM stock, pH 7.2). 2.0 μL of the protein sample was applied to 300 mesh Cu Quantifoil 1.2/1.3 grids (Quantifoil Microtools) that were glow discharged for 20 seconds at 25 mA....”

Was whatman paper treated w EGTA or ashless?

We did not pre-treat the filter papers with EGTA or any other chelators, and the filter paper is ashless. We changed this section as stated below to clarify. The catalog number for the filter papers are also stated.

Line 298 on page 13: “...The grids were blotted for 7 s at force 10 using single-layer Whatman ashless filter papers (Cat. #: 1442-055, GE Healthcare) and were plunged into liquid ethane using an FEI MarkIV Vitrobot at 8 °C and 100% humidity. The filter papers were not pre-treated with Ca²⁺ chelators or any other chemicals...”

How was the free ca²⁺ concentration in solution assessed?

Free Ca²⁺ concentration was calculated using Maxchelator calculator ([WEBMAXC STANDARD \(ucdavis.edu\)](http://WEBMAXC.STANDARD.ucdavis.edu)). This is now stated in the first paragraph of the results section.

Line 86 on page 4: “...Although the free Ca²⁺ concentration was calculated around 100 nM under these conditions using Maxchelator²⁰, the actual free Ca²⁺ concentration may be higher due to potential leakage of Ca²⁺ during the cryo-grid preparation as mentioned above...”

Reviewers' Comments:

Reviewer #1:

Remarks to the Author:

The authors answered all my comments and adjusted the manuscript accordingly. I am happy with their current version and do not have further remarks. I have only one minor remark. Authors can consider adding a few words to the discussion regarding so low open probability of the channel. Is it something expected and known from electrophysiology or it is an artifact of protein purification?

Reviewer #2:

Remarks to the Author:

The authors have done thorough rewriting and re-analysis of the structures, and have addressed my previous concerns and suggestions. I only have one minor comment, which I trust the authors can take care of without the need for re-review: In any of the figures showing maps, please indicate which exact map is being shown (i.e. overall map or locally refined map after symmetry expansion).

Reviewer #3:

Remarks to the Author:

Overall, the authors addressed most of the reviewers' remarks in this significantly improved manuscript. The cryo-EM data was greatly improved and supports the conclusions of the authors.

Minor comments: In supplementary figure 6b, masks are overlaid over a very partial map, so we don't see exactly what was masked. If a domain does not show up when the map is filtered to a given resolution, it often means that the resolution of the map at the location of this domain is much lower. The map should therefore be filtered to that resolution for the domain to become visible. The figure should be updated to reflect this.

This also applies to the loose maps in supp figure 1. They should be filtered to a resolution where the loose domains appear so that readers don't confuse them for maps with missing domains, which is not the case.

The addition of supplementary figure 8 is a welcome addition. A panel showing the density overlaid onto the model would help the readers assess the accuracy of the fit and whether indeed the resolution is too low to assess the presence of calcium.

I am still not convinced by the sequential attribution of states pre-A->pre-B->pre-C as there is not much data supporting it, but this minor point does not affect the overall validity of the story.

RESPONSE TO REVIEWERS:

We thank the reviewers for critically re-evaluating our recent manuscript and are delighted that our revised manuscript addressed most of their comments. We have updated the manuscript and figures to reflect the remaining changes requested.

Please find our point by point response (colored red) to the reviewers' comments directly underneath each comment below. The changes in our manuscript can also be tracked in our resubmitted document.

Reviewer #1 (Remarks to the Author):

The authors answered all my comments and adjusted the manuscript accordingly. I am happy with their current version and do not have further remarks.

Thank you!

I have only one minor remark. Authors can consider adding a few words to the discussion regarding so low open probability of the channel. Is it something expected and known from electrophysiology or it is an artifact of protein purification?

This is an important but difficult question to answer since it is often difficult to correlate the distribution of the particles after 3D classification and the distribution of the particles in the sample before cryo-grid preparation. Some factors affecting this discrepancy may involve a difference in the stability of the particles among the functional states during sample freezing and biased preference for particles in different functional states to be positioned in the imaged (hole) or unimaged (carbon) regions of the cryo-grids. Potential artifacts of protein purification may also contribute. It is also not possible to know the exact concentration of free Ca^{2+} concentration in the sample, which directly affects the open probability. Therefore, we respectfully decided to refrain from making any speculations regarding the particle distribution in our analysis and their correlation with the particle distribution in the sample.

Reviewer #2 (Remarks to the Author):

The authors have done thorough rewriting and re-analysis of the structures, and have addressed my previous concerns and suggestions.

Thank you!

I only have one minor comment, which I trust the authors can take care of without the need for re-review: In any of the figures showing maps, please indicate which exact map is being shown (i.e. overall map or locally refined map after symmetry expansion).

We indicated which map we used in all the figures involving maps.

Reviewer #3 (Remarks to the Author):

Overall, the authors addressed most of the reviewers' remarks in this significantly improved manuscript. The cryo-EM data was greatly improved and supports the conclusions of the authors.

Thank you!

Minor comments: In supplementary figure 6b, masks are overlaid over a very partial map, so we don't see exactly what was masked. If a domain does not show up when the map is filtered to a given resolution, it often means that the resolution of the map at the location of this domain is much lower. The map should therefore be filtered to that resolution for the domain to become visible. The figure should be updated to reflect this.

We performed local filtering using CryoSparc to low-pass the map for the inactive structure by local resolution and obtained a map that shows previously invisible domains. We used this map to update Supp. Fig 6.

This also applies to the loose maps in supp figure 1. They should be filtered to a resolution where the loose domains appear so that readers don't confuse them for maps with missing domains, which is not the case.

We added a panel showing the locally filtered map of the inactive structure mentioned in response to the previous comment to the Supp. Fig. 1.

The addition of supplementary figure 8 is a welcome addition. A panel showing the density overlaid onto the model would help the readers assess the accuracy of the fit and whether indeed the resolution is too low to assess the presence of calcium.

We modified the figure by adding another panel (Supp. Fig. 8d) showing the density map after local refinement and the modeled protein. The figure shows that the density does not reveal side chain positions around the Ca²⁺ binding site, although the domains can be placed into the density with high accuracy.

I am still not convinced by the sequential attribution of states pre-A->pre-B->pre-C as there is not much data supporting it, but this minor point does not affect the overall validity of the story.

To clarify the possibility of alternative transitions between the states, we modified the sentence that describes this transition as follows;

“.....The pre-active B and C structures adopt distinct conformations that are intermediates between the pre-active A and open state structures. Based on these conformational changes, we propose a sequential transition from pre-active A to B, then C, although the alternative transitions cannot be ruled out entirely. During the transition to the pre-active B state,.....”